# Association between recent COVID-19 diagnosis, depression and anxiety symptoms among slum residents in Kampala, Uganda

Solomon T. Wafula[1,2]*, Lesley L. Ninsiima[1], Hilbert Mendoza[1,3], John C. Ssempebwa[1], Florian Walter[4], David Musoke[1]

1 Department of Disease Control and Environmental Health, Makerere University School of Public Health, Kampala, Uganda, 2 Department of Infectious Disease Epidemiology, Bernhard Nocht Institute for Tropical Medicine, Hamburg, Germany, 3 Social Epidemiology and Health Policy, Department of Family Medicine and Population Health, University of Antwerp, Antwerp, Belgium, 4 Division of Nursing, Midwifery and Social Work, School of Health Sciences, Faculty of Biology, Medicine and Health, The University of Manchester, Manchester, United Kingdom

* swafula@musph.ac.ug

**Data Availability Statement:** All relevant data are within the paper and its Supporting Information file.

## Abstract

### Background

Despite the known link between poor living conditions and mental health, there has been little research on the mental health of slum dwellers worldwide. Although the Coronavirus disease 2019 (COVID-19) pandemic has led to an increase in mental health issues, little focus has been given to the impact on slum dwellers. The study aimed to investigate the association between recent COVID-19 diagnosis and the risk of depression and anxiety symptoms among people living in an urban slum in Uganda.

### Methods

A cross-sectional study was conducted among 284 adults (at least 18 years of age) in a slum settlement in Kampala, Uganda between April and May 2022. We assessed depression symptoms and anxiety using validated Patient Health Questionnaire (PHQ-9) and Generalized Anxiety Disorder assessment tool (GAD-7) questionnaires respectively. We collected data on sociodemographic characteristics, and self-reported recent COVID-19 diagnosis (in the previous 30 days). Using a modified Poisson regression, adjusted for age, sex, gender and household income, we separately provided prevalence ratios and 95% confidence intervals for the associations between recent COVID-19 diagnosis and depressive and anxiety symptoms.

### Results

Overall, 33.8% and 13.4% of the participants met the depression and generalized anxiety screening criteria respectively and 11.3% were reportedly diagnosed with COVID-19 in the previous 30 days. People with recent COVID-19 diagnosis were more likely to be depressed (53.1%) than those with no recent diagnosis (31.4%) (p<0.001). Participants who were recently diagnosed with COVID-19 reported higher prevalence of anxiety (34.4%) compared

**Funding:** This work was supported by Makerere University School of Public Health under the Small Grants Programme (MakSPH-GRCB/18-19/01/02 to STW). The funders had no role in study design, data collection and analysis, decision to publish, or preparation of the manuscript.

**Competing interests:** The authors have declared that no competing interests exist.

to those with no recent diagnosis of COVID-19 (10.7%) (p = 0.014). After adjusting for confounding, recent diagnosis with COVID-19 was associated with depression (PR = 1.60, 95% CI 1.09–2.34) and anxiety (PR = 2.83, 95% CI 1.50–5.31).

## Conclusion

This study suggests an increased risk of depressive symptoms and GAD in adults following a COVID-19 diagnosis. We recommend additional mental health support for recently diagnosed persons. The long-term of COVID-19 on mental health effects also need to be investigated.

## Introduction

In 2020, the World Health Organization (WHO) declared Coronavirus disease 2019 (COVID-19) a pandemic and has since caused detrimental effects on people's health and well-being globally. By February 2023, there have been more than 750 million cases and 6.8 million deaths [1, 2]. The uncertainties and unpredictable nature of the pandemic resulted in individuals experiencing various forms of mental health problems such as depression, anxiety, psychosocial dysfunction, dissociation disorders, substance abuse, and insomnia. It was projected that these effects would be most severe in low-resourced settings due to constraints related to weak health systems with limited access to mental health services [3].

Low-resourced settings such as slum residences comprise informal dwelling units which are poorly constructed, crowded with poor sanitary conditions and environmental pollution [4, 5]. Slum dwellers usually have poor physical and economic access to health care including mental health care [6]. Most informal settlements comprise an assortment of individuals (e.g., low-income earners, asylum seekers, refugees), who are at high risk of mental health problems due to multiple stressors and this may be further aggravated by the COVID-19 pandemic [3]. Due to unfavourable conditions of unstable income sources, high dependence on daily casual jobs, and threatened livelihoods and food security [7, 8], slum dwellers may be at elevated risk of experiencing mental health challenges. A constellation of individual, social and cultural factors that contribute to psychological distress in informal settings have been highlighted and include trauma, violence, community insecurity, and healthcare barriers [9]. Mental health among informal settlement dwellers is still limited, including both before and during the COVID-19 pandemic [10]. Available evidence suggests that psychological distress during the pandemic has been high in various parts of Africa, For example, in a phone-based study in urban areas of Burkina Faso, Ethiopia, and Egypt, 28% of participants had symptoms of mild, moderate, or severe psychological distress [11]. High prevenance of mental health disorders have been reported in East Africa [12], Mali [13], south Africa [14, 15], Ghana [16] etc but most of these were conducted among health care workers or the general population.

Evidence from recent epidemics such as SARS, MERS and Ebola poignantly illustrate the psychosocial impacts of the epidemic and related restrictions [17, 18]. During these epidemics, the fear of infection or death, helplessness, depression, anxiety, social isolation, and stigma are the most reported psychological manifestations [19]. For example, approximately 15% of individuals quarantined due to SARS epidemic in the Canadian city of Toronto showed signs of depression and post-traumatic stress disorder [20]. Economic-related mental distress has been reported to have a negative impact on COVID-19 vaccine uptake [21]. Similarly, higher prevalence of psychological effects such as fear, anxiety and depression were reported during the 2013–2016 West Africa Ebola outbreak [22]. Whereas several studies on the effect of COVID-

19 on mental health have been conducted in other populations and settings such as hospitals [23–26], there was limited evidence regarding the effect of recent diagnosis with COVID-19 on depression and anxiety, especially among socioeconomically deprived populations such as slum dwellers. Preliminary evidence suggests that symptoms of anxiety and depression range from 16 to 28% and self-reported stress is around 8% during COVID-19 pandemic throughout the world [27]. To date, little is known about the burden of mental health conditions disease in urban slums and related research among slum residents in Uganda and globally [10]. We, therefore, set out to investigate the association between recent COVID-19 diagnosis and depressive and anxiety symptoms among residents in an urban slum in Uganda.

## Methods

### Study design and setting

This study used cross-sectional, population-based data from adults (at least 18 years of age) residing in Bwaise, an urban slum setting in Kampala, Uganda from April to May 2022. Bwaise is one of Kampala's most densely populated slums, distinguished by mostly informal and sub-standard housing and small-scale businesses. It has a high population density, crowded households, and low socioeconomic status. About 60% of the urban population in Kampala city live in areas that are defined as informal. The average household size is between 5–10 people, who typically share very tiny spaces like one or two-roomed houses [28]. This makes preventive measures like social distancing extremely difficult to abide by. Between 3 January 2020 and 11 February 2022, Ugandan's reported cumulative COVID-19 cases reached 162,639, with 3,575 deaths [2]. In 2018, there were a total of 6,937 health facilities in Uganda, and private providers owned 55% of them. Private providers have a dominant presence in urban areas, particularly in the capital city of Kampala where they constitute 99% of the facilities [29]. The eligible study population consisted of adults (18 years and above) from households who had resided in the area for at least three months, and would provide informed consent. Those who had severe mental illnesses or declined consent were excluded.

### Sample size and sampling procedures

A sample of 284 adults from households in the Bwaise slum participated in the study. This is a secondary analysis of the main study which assessed the effect of indoor air quality on health outcomes. The sample size calculated was for the main study based on the prevalence of respiratory problems (main outcome)(p) at 19.4%, 80% power, 5% power and 15% non-response rate but this sample size still had acceptable power for this secondary objective [30]. A representative sample of households were selected from the three parishes in Bwaise. Within each parish, one zone was selected randomly by ballot method, and at least 93 households were selected from each zone using a systematic sampling approach, with a sampling interval obtained by dividing the number of households (based on records from the Zonal local council office) by the number of households needed in each zone (i.e., by dividing the total number of households by 93). Sampling and interviews started at the local council office and then north direction, then clockwise until the target sample size was realised for each zone. A selected household was replaced by the next household if the original household had no eligible respondents or did not consent.

### Study procedures and measures

We organized face-to-face questionnaire interviews with participants through a consenting process by experienced and trained research assistants. Research assistants with a background

training in health sciences captured participant responses on KoboCollect; a mobile data collection app (https://www.kobotoolbox.org/). To gather exposure data, we obtained information on two exposure aspects: (1) Recent covid-19 diagnosis defined as whether the participants had been diagnosed with COVID-19 in the past 30 days, and this was self-reported, and (2) COVID-19 experience; whether they had any personal experiences with COVID-19 in the past 30 days, such as a positive diagnosis, caring for someone with COVID-19, or experiencing the death of a family member or close friend due to COVID-19. The primary outcomes were current depression symptoms and generalized anxiety disorder (within the preceding 2 weeks) as per the standard practice. Depressive symptoms were assessed using a validated 9-item Patient Health Questionnaire (PHQ-9) [31], with each item scored as 0 (not at all), 1 (several days), 2 (more than half the days) and 3 (nearly every day). The PHQ-9 is a commonly used tool for assessing depression in individuals, comprising nine questions that are rated on a four-point scale ranging from 0 (not at all) to 3 (nearly every day) based on their experiences over the past two weeks." Thus, the total score of the PHQ-9 can range from 0 to 27. As previously recommended [32], we used the cutoff score of 10, corresponding with at least a moderate level of depression. Anxiety was assessed using a 7-item validated Generalized Anxiety Disorder (GAD-7) tool to screen the presence of generalized anxiety disorder and assess its severity [33]. Each question in the GAD-7 tool is scored on a four-point Likert scale (from "0 = not at all" to "3 = almost every day") with total scores ranging from 0 to 21 and is obtained by summing the raw scores of the seven items. The recommended screening cutoff was ≥10, corresponding with at least a moderate anxiety level. "The Cronbach's alpha coefficient for PHQ-9 and GAD-7 in the current study was 0.71 and 0.80, which suggests high reliability.

Covariates: We collected data on covariates and potential confounding variables including the socio-demographic characteristics (age in complete years, sex (male, female) marital status (single, separated/divorced, married), household income in Uganda shillings, education level (none, primary, post-primary), duration of stay in the area in complete years, having under-five children (Yes, no)). Participants were also asked if they used the following psychoactive substances on more than one occasion in the last 30 days (the substances were alcohol, smoking, and marijuana). Data collection tools were developed following a critical review of the existing literature [34–37]. These tools were pretested in another slum setting in Kampala and later revised to ensure minimize ambiguity and ensure understanding.

## Data management and analysis

We created derived variables for two factors: the length of the stay and household income. Firstly, we created a categorical variable for length of stay by dividing the years into three categories based on five-year intervals: (i) less than or equal to 5 years, (ii) 6 to 10 years, and (iii) more than 10 years. Secondly, we converted the household income in Uganda shillings to US Dollars (I USD = 3500 UGX) and categorized it as follows: (i) less than 50 USD, (ii) between 50 USD to 150 USD, and (iii) greater than 150 USD. We used the median national monthly household income of around 50 USD as the cut-off for the first category, while for the second category, we used the median monthly income of approximately 150 USD for the capital, Kampala) [38].

Statistical analyses were performed using Stata 14 (Stata Corp, Texas, USA). Descriptive statistics such as frequencies and percentages were used for categorical data. For continuous data, we used medians and interquartile ranges. Participant characteristics by depression and anxiety symptoms status were compared using the Wilcoxon rank sum test for non-normally distributed continuous variables and the Chi-squared test for dichotomous variables. We also

performed two separate multivariable modified Poisson regressions adjusting for age, sex, gender and household income [39] and producing prevalence ratios (PRs) and corresponding 95% confidence interval (95% CI) for the association between recent COVID-19 infection and depression and anxiety. All statistical tests were two-tailed, and statistical significance was assumed when a p-value was $\leq 0.05$.

## Results

### Background characteristics of participants

In total, 284 subjects participated in the study including responding to the PHQ-9 and GAD-7 questionnaires. Characteristics of all participants are described for depressive and anxiety disorders determined by PHQ-9 and GAD-7 cutoffs $\geq 10$. Most participants were females 85.2% and had a median age of 29 years (IQR = 24,37). The median depression score based on PHQ-9 was 8 (6,10) while for anxiety, the median score was 6 (4,8) based on the GAD-7. The prevalence of depression was 33.8% and anxiety was 13.4%, which differed significantly depending on income (p<0.001) and recent positive COVID-19 diagnosis status (p<0.001) (Table 1).

### Association between recent COVID-19 diagnosis and depressive symptoms

In unadjusted analysis, there was a significant association between recent SARS-COV-2 infection with household income and smoking in households. After adjusting for age, gender and income, recent COVID-19 diagnosis was positively associated with current depressive symptoms (PR = 1.60, 95% CI 1.09–2.34) (Table 2).

### Association between recent COVID-19 diagnosis and generalized anxiety disorder

In adjusted analysis, participants who had been diagnosed with COVID-19 within 30 days of the survey had a 2.8 times higher risk of anxiety compared to their counterparts (PR = 2.83, 95%CI 1.50–5.31). Higher monthly incomes (i.e. USD 50–150 (PR = 0.38, 95%CI (0.19–0.76) and USD> 150 (PR = 0.30, 95%CI (0.11–0.87)) were associated with lower risk of anxiety symptoms compared to those who earned less than USD 50. The likelihood of meeting anxiety screening criteria was 2.23 times higher among participants who reported regularly drinking alcohol compared to those with no/limited alcohol consumption (PR = 2.23, 95CI 1.12–4.42) (Table 3).

## Discussion

In this community-based study, we examined the prevalence of depressive symptoms and anxiety and their association with recent COVID-19 diagnosis among adult slum dwellers in Kampala, Uganda. Our findings highlight the challenges that individuals affected by COVID-19 face, and are consistent with the larger literature on the impact of COVID-19 on mental health in the general population. In this study, a recent COVID-19 diagnosis was associated with both depression and anxiety. For instance, it increased the likelihood of depressive symptoms by 60%. For Generalized anxiety disorder (GAD), the prevalence was 2.8 times higher than for those who did not have a recent infection after adjusting for age, gender, household income and use of psychoactive substances.

Our study results revealed a high prevalence of anxiety and depression among participants (13.4% and 33.8%, respectively). High levels of depression and anxiety raise concerns about the capacity of mental healthcare systems to respond during the pandemic. The prevalence of depression in among slum dwellers in this study was higher than national pre-pandemic levels

**Table 1. Characteristics of household participants.**

| Characteristics | Overall | Current Depressive symptoms | | | Generalized anxiety disorder | | |
|---|---|---|---|---|---|---|---|
| | | No | Yes | p-value | No | Yes | p-value |
| **Total** | **284** | 188 (66.2) | 96 (33.8) | | 246 (86.6) | 38 (13.4) | |
| **Gender** | | | | | | | |
| Female | 242 (85.2) | 158 (65.3) | 84 (34.7) | 0.438 | 207 (85.5) | 35 (14.5) | 0.198 |
| Male | 42 (14.8) | 30 (71.4) | 12 (28.6) | | 39 (92.9) | 3 (7.1) | |
| **Age of respondents (in years)** | | | | | | | |
| Median (IQR) | 29 (24, 37) | 29 (25,36) | 29 (23,38) | 0.689[1] | 29 (24, 37) | 29 (23, 35) | 0.924[1] |
| < 30 | 146 (51.4) | 97 (66.4) | 49 (33.6) | 0.736 | 126 (86.3) | 20 (13.7) | 0.794 |
| 30–45 | 108 (38.0) | 73 (67.6) | 35 (32.4) | | 95 (88.0) | 13 (12.0) | |
| > 45 | 30 (10.6) | 18 (60.0) | 12 (40.0) | | 25 (83.3) | 5 (16.7) | |
| **Marital status** | | | | | | | |
| Married or living with partner | 136 (47.9) | 93 (68.4) | 43 (31.6) | 0.161 | 120 (88.2) | 16 (11.8) | |
| Separated | 43 (15.1) | 23 (53.5) | 20 (46.5) | | 32 (74.4) | 11 (25.6) | |
| Single | 105 (37.0) | 72 (68.6) | 33 (31.4) | | 94 (89.5) | 11 (10.5) | |
| **Education level** | | | | | | | |
| No formal education | 25 (8.8) | 16 (64.0) | 9 (36.0) | 0.201 | 20 (80.0) | 5 (20.0) | 0.514 |
| Primary | 114 (40.1) | 69 (60.5) | 45 (39.5) | | 98 (86.0) | 16 (14.0) | |
| Post primary | 145 (51.1) | 103 (71.0) | 42 (29.0) | | 128 (88.3) | 17 (.7) | |
| **Occupation** | | | | | | | |
| Employed | 58 (20.4) | 131 (67.2) | 64 (32.8) | 0.490 | 169 (86.7) | 26 (13.3) | 0.973 |
| Unemployed | 72 (25.4) | 48 (66.7) | 24 (33.3) | | 62 (86.1) | 10 (13.9) | |
| Other | 17 (6.0) | 9 (52.9) | 8 (47.1) | | 15 (88.2) | 2 (11.8) | |
| **Owner of the dwelling** | | | | | | | |
| Yes | 26 (9.2) | 19 (73.1) | 7 (26.9) | 0.437 | 23 (88.5) | 3 (11.5) | 0.772 |
| No | 258 (90.8) | 169 (65.5) | 89 (34.5) | | 223 (86.4) | 35 (13.6) | |
| **Household income (USD)** | | | | | | | |
| Median (IQR) | 83.3 (41.7, 125) | 83.3 (55.5,125) | 83.3 (41.7,125) | 0.061[1] | 83.3 (58.3, 125) | 41.6 (22.2, 83.3) | <0.001[1] |
| < 50 | 77 27.1 | 44 (57.1) | 33 (42.9) | 0.122 | 55 (71.4) | 22 (28.6) | < 0.001 |
| 50–150 | 163 57.4 | 115 (70.6) | 48 (29.5) | | 150 (92.0) | 13 (8.0) | |
| > 150 | 44 15.5 | 29 (65.9) | 15 (34.1) | | 41 (93.2) | 3 (6.8) | |
| **Duration of residence (years)** | | | | | | | |
| < = 5 | 148 (52.1) | 96 (64.9) | 52 (35.1) | 0.218 | 125 (84.5) | 23 (15.5) | 0.366 |
| 6–10 | 57 (20.1) | 34 (59.7) | 23 (40.3) | | 49 (86.0) | 8 (14.0) | |
| > 10 | 79 (27.8) | 58 (73.4) | 21 (26.4) | | 72 (91.1) | 7 (8.9) | |
| **COVID-19** | | | | | | | |
| **Diagnosed with COVID-19 in last 30 days: No** | 252 (88.7) | 173 (68.7) | 79 (31.4) | 0.014 | 225 (89.3) | 27 (10.7) | <0.001 |
| Yes | 32 (11.3) | 15 (46.9) | 17 (53.1) | | 21 (65.6) | 11 (34.4) | |
| **Had family member or close friend who suffered or died from COVID-19 in the previous 30 days: No** | 220 (77.5) | 147 (66.8) | 73 (33.2) | 0.682 | 188 (85.5) | 32 (14.6) | 0.285 |
| yes | 64 (22.5) | 41 (64.1) | 23 (35.9) | | 58 (90.6) | 6 (9.4) | |
| **Substance use in last 30 days** | | | | | | | |
| Alcohol use: No | 175 (61.6) | 120 (68.6) | 55 (31.4) | 0.284 | 159 (90.9) | 16 (9.1) | 0.008 |
| Yes | 109 (38.4) | 68 (62.4) | 41 (37.6) | | 87 (79.8) | 22 (20.2) | |
| Smoking: No | 243 (85.6) | 166 (68.3) | 77 (31.7) | 0.067 | 215 (88.5) | 28 (11.5) | 0.025 |
| Yes | 41 (14.4) | 22 (53.7) | 19 (46.3) | | 31 (75.6) | 10 (24.4) | |
| Marijuana: No | 265 (93.3) | 176 (66.4) | 89 (33.6) | 0.772 | 229 (86.4) | 36 (13.6) | 0.705 |

*(Continued)*

**Table 1.** (Continued)

| Characteristics | Overall | Current Depressive symptoms | | | Generalized anxiety disorder | | |
|---|---|---|---|---|---|---|---|
| | | No | Yes | p-value | No | Yes | p-value |
| Yes | 19 (6.7) | 12 (63.2) | 7 (36.8) | | 17 (89.5) | 2 (10.5) | |
| **Have children under 5 years: No** | 128 (45.1) | 86 (67.2) | 42 (32.8) | 0.749 | 117 (91.4) | 11 (8.6) | 0.032 |
| Yes | 156 (54.9) | 102 (65.4) | 54 (34.6) | | 129 (82.7) | 27 (17.3) | |

[1] p-value based on the Wilcoxon rank sum test; other p-values based on Pearson chi-square statistics.

**Table 2. Association between recent COVID-19 diagnosis and depressive symptoms.**

| Attributes | Current depressive symptoms | | | |
|---|---|---|---|---|
| | Unadjusted analysis | | Adjusted analysis | |
| | PR (95% CI) | p-value | PR (95% CI) | p-value |
| *Sociodemographic characteristics* | | | | |
| **Gender** | | | | |
| Female | 1 | | | |
| Male | 0.82 (0.45–1.51) | 0.454 | 0.83 (0.49–1.39) | 0.470 |
| **Age of respondents (in years)** | 1.00 (0.98–1.02) | 0.872 | 1.00 (0.98–1.01) | 0.896 |
| **Marital status** | | | | |
| Married or with partner | 1 | | 1 | |
| Separated | 1.47 (0.86–2.50) [+] | 0.062 | 1.25 (0.81–1.92) | 0.312 |
| Single | 0.99 (0.63–1.56) | 0.975 | 0.92 (0.62–1.36) | 0.665 |
| **Education level** | | | | |
| No formal education | 1 | | | |
| Primary | 1.09 (0.54–2.24) | 0.752 | | |
| Post primary | 0.80 (0.39–1.65) | 0.464 | | |
| **Occupation** | | | | |
| Employed | 1 | | 1 | |
| Unemployed | 1.01 (0.69–1.49) | 0.937 | 0.98 (0.65–1.48) | 0.931 |
| Other | 1.43 (0.83–2.47) | 0.194 | 1.41 (0.78–2.55) | 0.253 |
| **Owner of the dwelling:** | 1.28 (0.59–2.76) | 0.459 | | |
| **Length of stay (years)** | | | | |
| < 5 | 1 | | | |
| 5–10 | 1.14 (0.70–1.88) | 0.481 | | |
| >10 | 0.76 (0.46–1.26) | 0.201 | | |
| **Household income (USD)** | | | | |
| < 50 | 1 | | 1 | |
| 50–150 | 0.69 (0.48–0.98) | **0.036** | 0.80 (0.55–1.15) | 0.224 |
| > 150 | 0.80 (0.49–1.29) | 0.356 | 0.83 (0.51–1.36) | 0.457 |
| *COVID-19 experiences* | | | | |
| Diagnosed with COVID-19 in the last 30 days | 1.69 (1.01–2.86) * | **0.006** | 1.60 (1.09–2.34) | **0.016** |
| Family or close friend suffered or died from COVID-19 in the previous 30 days | 1.08 (0.67–1.73) | 0.679 | | |
| *Psychoactive substance uses* | | | | |
| Alcohol use | 1.20 (0.80–1.79) | 0.281 | | |
| Regular smoking | 1.46 (0.89–2.42) | **0.049** | 1.29 (0.85–1.96) | 0.225 |
| Marijuana or cannabis use | 1.09 (0.51–2.36) | 0.768 | | |
| **Have children under 5** | 1.05 (0.70–1.58) | 0.750 | | |

**Table 3. Association between recent COVID-19 diagnosis and anxiety disorder.**

| Attributes | Generalized Anxiety disorder | | | |
|---|---|---|---|---|
| | Unadjusted analysis | | Adjusted analysis | |
| | PR (95% CI) | p-value | PR (95% CI) | p-value |
| *Sociodemographic characteristics* | | | | |
| **Gender** | | | | |
| Female | 1 | | 1 | |
| Male | 0.49 (0.16–1.54) | 0.223 | 0.57 (0.21–1.52) | 0.260 |
| **Age of respondents (in years)** | 1.01 (0.98–1.03) | 0.592 | 1.02 (0.98–1.05) | 0.347 |
| **Marital status** | | | | |
| Married or with partner | 1 | | 1 | |
| Separated | 2.17 (1.01–4.68) | **0.027** | 1.56 (0.72–3.42) | 0.262 |
| Single | 0.89 (0.41–1.92) | 0.754 | 0.82 (0.41–1.63) | 0.567 |
| **Education level** | | | | |
| No formal education | 1 | | | |
| Primary | 0.70 (0.26–1.92) | 0.444 | | |
| Post primary | 0.59 (0.22–1-.59) | 0.247 | | |
| **Occupation** | | | | |
| Formal or informal employment | 1 | | | |
| Unemployed | 1.04 (0.52–2.05) | 0.906 | | |
| Other | 0.88 (0.22–3.41) | 0.856 | | |
| **Owner of the dwelling: No** | 1.18 (0.36–3.82) | 0.775 | | |
| Length of stay | | | | |
| < 5 years | 1 | | 1 | |
| 5–10 | 0.90 (0.41–2.01) | 0.789 | 1.14 (0.53–2.44) | 0.733 |
| >10 | 0.57 (0.24–1.33) | 0.170 | 0.53 (0.21–1.33) | 0.176 |
| **Household income (USD)** | | | | |
| < 50 | 1 | | | |
| 50–150 | 0.28 (0.14–0.52) | **0.001** | 0.38 (0.19–0.76) | **0.006** |
| > 150 | 0.24 (0.08–0.75) | **0.015** | 0.30 (0.11–0.87) | **0.026** |
| *COVID-19 experiences* | | | | |
| **Diagnosed with COVID-19 in the last 30 days** | 3.21 (1.76–5.83) | **<0.001** | 2.83 (1.50–5.31) | **0.001** |
| **Family or close friend suffered or died from COVID-19 in the previous 30 days** | 0.64 (0.27–1.54) | 0.298 | | |
| *Psychoactive substance use in the last 30 days* | | | | |
| Alcohol | 2.21 (1.16–4.20) | **0.010** | 2.23 (1.12–4.42) | **0.022** |
| Smoking | 2.12 (1.03–4.35) | **0.022** | 1.21 (0.52–2.85) | 0.662 |
| Marijuana | 0.77 (0.19–3.22) | 0.711 | | |
| **Have children under 5** | 2.01 (1.00–4.06) | **0.038** | 1.93 (0.98–3.82) | 0.058 |

of 29.3% [40]. However, it's important to note that the national survey had a larger sample and was not specifically focused on slum dwellers. Studies conducted in the general adult population of Uganda have shown that mental health was significantly impacted by the COVID-19 pandemic, even among those who were not directly infected [23, 26]. In one a study [23], the prevalence rates were reported to be in excess of 80% for depression and 90% for anxiety. Although outcome measures are not directly comparable to our study, the reported rates appear worryingly high. As noted in that study, several limitations related to the data collection might have artificially increased the prevalence rates. However, the study underlines the significance of the mental health impact attributed to COVID-19 and related restrictions. Our findings support new evidence from other low and middle-income countries (LMICs) on the state

of mental health during the pandemic. For instance, a recent study conducted in a socially and economically comparable setting in Kenya, reported elevated rates of depression (34.1%) and GAD (14.0%) [3] while in Bangladesh, the estimated prevalence of depression was 33.0% [41]. A further study from the United Kingdom [42] has compared pre- and post-COVID-19 rates of psychological distress and has shown that deterioration of mental health did not recover after COVID-19 restrictions were eased. Further evidence indicates mental health problems among adults, especially during a health crisis, impact negatively on child developmental and behavioural outcomes [43]. The implication is that children of parents/guardians with such environments and backgrounds may be at elevated risk to experience various cognitive, behavioural, and emotional problems later in life if the problem is not addressed [44].

Although several studies have highlighted the elevated risk of mental health disorders and psychological distress during the COVID-19 pandemic [15, 25, 45], there seems to be limited research focusing specifically on how being recently diagnosed with COVID-19 affects individuals' mental health. In our study, recently diagnosed (< 30 days) participants were 1.5 times and 2.8 times more likely to be diagnosed to have depression and anxiety respectively compared to those who did not have COVID-19 diagnosis within the previous 30 days. In line with these findings, evidence from the United States [23] showed that experiencing a COVID-19 infection was associated with an increased risk of common mental disorders such as anxiety and depression in the acute phase of the infection. This is also in agreement with multiple other studies reporting an elevation in mental health problems following infection [45–48]. To our knowledge, our study is one of the first to investigate the influence of recent COVID-19 diagnosis on depression and anxiety in Uganda. Our findings reinforce the fact that the month following an infection seems to be associated more strongly with depressive symptoms and anxiety than experiencing a COVID-19 infection at any point in the past [49]. A possible explanation for these findings is that additional stressors related to the impact of COVID-19 infections such as financial problems, isolation from others, grief and worries about others could contribute to the experience of common mental health problems. However, studies have suggested that underlying biological mechanisms could also explain the increase in the prevalence of mental disorders [50, 51]. It is well-established that the mechanisms, predominantly driven by the body's immune response and associated cytokine activity can affect the neurological structures of patients [52]. It is therefore reasonable to suggest that these changes could contribute to the observed effects on mental health. However, these mechanisms are not fully understood and require further research.

## Strengths and limitations

The study has several strengths: Firstly, the study focuses on a specific population of slum residents. This allows for a more targeted approach to understanding the impact of COVID-19 on mental health, rather than a more general study that may not be as relevant to this specific population. Secondly, the study used standardized measures to assess depression and anxiety symptoms. These measures can help ensure that the results are reliable and valid, making it easier to draw conclusions about the relationship between COVID-19 diagnosis and mental health outcomes. Although this is one of the first studies to examine the association between recent COVID-19 diagnosis and depressive symptoms and GAD in an informal settlement in Uganda, it had several limitations. First of all, a recent COVID-19 diagnosis was self-reported (as told to participants by medical practitioners after tests) and we could not validate this. It is possible that participants misclassified themselves as not having a COVID-19 infection because they were not tested or diagnosed during the study period (especially if they were asymptomatic). Second, we only collected data on a limited set of confounding variables,

which, coupled with the relatively small sample size, may have hampered the robustness of our findings because we were unable to account for all relevant confounders. Thirdly, we evaluated mental health at a single point in time, which restricts our capacity to examine changes in the occurrence of mental health issues over time. It also prevented us from determining whether the relationships between COVID-19 diagnosis and mental health changed as time passed after diagnosis or as the pandemic progressed. However, considering that our findings correspond with the wider literature, we are confident that despite unmeasured confounding, this association would still hold even if we were to adjust for further covariates. Longitudinal studies may be needed in future to assess the effect of COVID-19 diagnosis on mental health over time.

## Conclusion

Overall, the study suggests an elevated risk for depressive and anxiety symptoms among adults who had a recent COVID-19 diagnosis (within the previous 30 days compared to those who had not been diagnosed. The findings of this study have several implications. First, it highlights the need for mental health services and support for individuals who have been diagnosed with COVID-19, particularly in resource-limited settings such as slums in Kampala. Additionally, the study suggests the need for additional support to address this burden and ensure that people receive mental /emotional treatment when necessary. While the impact of COVID-19 on mental health has been studied within 30 days of diagnosis, there may be longer-term effects that have not yet been fully understood. As a result, the authors recommend that further research should be conducted to evaluate the potential long-term consequences of COVID-19 infections on mental health. Overall, the study underscores the importance of addressing the mental health consequences of COVID-19, particularly in vulnerable populations.

## Supporting information

**S1 File.**
(DTA)

## Acknowledgments

We appreciate the local authorities for administrative support as well as the research assistants and the participants for without which this study would not have been possible.

## Ethical consideration

Ethical approval was obtained from Makerere University School of Public Health Higher Degrees Research and Ethics Committee (HDREC: Ref No. SPH-2021-99) and Uganda National Council of Science and Technology (UNCST: Ref No. SS996ES). The Kampala Capital City Áuthority gave permission for the study's implementation, provided advise and support with planning of study activities. We followed principles of the Helsinki declaration and written informed consent was obtained from each participant. Research Assistants followed all the COVID-19 risk mitigation protocols such as face masking, social distancing and hand washing during data collection to minimize risk of infection and transmission.

## Author Contributions

**Conceptualization:** Solomon T. Wafula, Hilbert Mendoza, John C. Ssempebwa, David Musoke.

**Data curation:** Solomon T. Wafula.

**Formal analysis:** Solomon T. Wafula.

**Funding acquisition:** Solomon T. Wafula.

**Investigation:** Solomon T. Wafula.

**Methodology:** Solomon T. Wafula, Florian Walter.

**Project administration:** Solomon T. Wafula, Lesley L. Ninsiima.

**Resources:** Solomon T. Wafula.

**Software:** Solomon T. Wafula.

**Supervision:** Solomon T. Wafula, Hilbert Mendoza.

**Writing – original draft:** Solomon T. Wafula, Lesley L. Ninsiima, Hilbert Mendoza, John C. Ssempebwa, Florian Walter, David Musoke.

**Writing – review & editing:** Solomon T. Wafula, Lesley L. Ninsiima, Hilbert Mendoza, John C. Ssempebwa, Florian Walter, David Musoke.

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
