## [Decision Letter · Decision Letter 0]

23 Feb 2023

PONE-D-22-35388Association between recent COVID-19 diagnosis on depression and anxiety symptoms among slum residents in Kampala, UgandaPLOS ONE

Dear Dr. Wafula,

Thank you for submitting your manuscript to PLOS ONE. After careful consideration, we feel that it has merit but does not fully meet PLOS ONE’s publication criteria as it currently stands. Therefore, we invite you to submit a revised version of the manuscript that addresses the points raised during the review process.

We look forward to receiving your revised manuscript.

Kind regards,

Seth Kwabena Amponsah, PhD

Academic Editor

PLOS ONE

Journal Requirements:

Additional Editor Comments:

Do well to address all comments raised by two Reviewers, point-by-point.

Reviewers' comments:

Reviewer's Responses to Questions

**Comments to the Author**

1. Is the manuscript technically sound, and do the data support the conclusions?

Reviewer #1: Yes

Reviewer #2: Yes

2. Has the statistical analysis been performed appropriately and rigorously? 

Reviewer #1: Yes

Reviewer #2: Yes

3. Have the authors made all data underlying the findings in their manuscript fully available?

Reviewer #1: Yes

Reviewer #2: Yes

4. Is the manuscript presented in an intelligible fashion and written in standard English?

Reviewer #1: Yes

Reviewer #2: Yes

5. Review Comments to the Author

Reviewer #1: Review comments

Title: Association between recent COVID-19 diagnosis on depression and anxiety symptoms among slum residents in Kampala, Uganda

Abstract

The background should mention briefly the rationale for choosing slum settlements.

The tools that were used in data collection especially for anxiety and depressive symptoms should be mentioned.

It will be important to indicate the incidence of covid-19 diagnosis within that past 30 days in the results

Introduction

Authors should provide references for sentence in lines 63 and 64

Generally, the introduction section is well written but authors should revise sections about low literature on COVID and slums since s cursory search online provides some evidence to guide the authors. Authors should focus building a case for their study based on what has been reported in relation to COVID and not so much on SARS etc.

Methods

Location of the study

Some detailed description of the study area should be provided. What is the population, facilities, healthcare access, COVID related infection rates etc should be provided since it’s one of Kampala's most densely populated slums and such information should be readily available.

Sampling

Line 87 - Sample size was calculated for the main study. It’s not clear what authors mean though I guess it’s been described earlier?

Procedure

How was COVID diagnosis confirmed?

How was the covid situation at the time of data collection and what considerations were made?

Ethics?

Information on covid protocol missing.

Data collection and measures

The PHQ and GAD have to be described well. How were the responses scored and categorized based on the scores for the items in the questionnaire?

How did the authors ensure the validity/reliability of the questionnaire?

Was the study pilot tested? What was the outcome of the pilot and what modifications were made afterwards?

Results

The results are well presented. A few comments though:

In the tables, there was information on substance use, income, covid experiences etc.

1. Each study variable has to be clearly indicated in the methods and how they were assessed.

2. What was the rationale for choosing the income categories?

Discussion.

I would rather authors stick to depressive and anxiety symptoms since the study tools and approach does not qualify for a diagnostic criterion for depression and anxiety to be used in the study.

The strengths of this study did not come out well.

Conclusion

Present a brief summary of major findings and include the implications of such findings to practice, policy, clinical care and the current COVID-19 pandemic.

Reviewer #2: In this manuscript, Wafula et al aimed to establish a link between COVID-19 and depression and generalized anxiety disorder among poor people in inner-city slums in Uganda. While their finding is of interest, especially around this time when COVID-19 still prevails, the manuscript cannot be accepted for publication in its current form due to several issues including missing punctuation marks and use of wrong punctuation marks, which require a major revision. Below are my comments which authors should address to help improve the quality of their manuscript:

TITLE

Authors should rephrase their title. They could either say "Association between recent COVID-19 diagnosis, depression and anxiety symptoms..." or "Effect of COVID-19 on depression and anxiety symptoms..."

ABSTRACT

Authors should write the full meaning of COVID-19, then put the abbreviation in parenthesis before using it in subsequent sentences. They should also write the full meaning of GAD. I believe it is generalized anxiety disorder. They should specify whether the 284 individuals recruited in the study were children, adults or both. Otherwise, they should state the minimum age of the study participants. In addition, authors should also use p-values to state whether or not significant difference exists the study groups.

INTRODUCTION

In line 38, authors should write the full meaning of COVID-19, then put the abbreviation in parenthesis before using it in subsequent sentences.

Among the detrimental effects of COVID-19 in line 39, authors should include a statement on statistics of COVID-19-associated mortality.

In line 46, authors should delete the preposition "of" that appeared after the word "comprise".

In line 61, authors should provide additional examples of the COVID-19-associated psychological distress beyond Burkina Faso, Ethiopia and Egypt.

In line 71, authors should replace the word "elsewhere" with a word or a phrase that defines the opposite of slum dwellers or slum settlement. This should be followed by a cited reference from published literature.

In line 79, authors should replace semi-colon (;) with a comma (,)

In line 80, authors should include a statement on why the study lasted from April to May, 2022

Did authors consider any inclusion and exclusion criteria in their study design? If yes, what were these criteria? If no, why?

In lines 86 and 87, authors should briefly state how they calculated sample size for the main study.

In line 89, authors should briefly explain what they mean by "systematic sampling"

What is the meaning of "LC" as used in line 91?

In line 92, authors should introduce a comma (,) after the word "direction"

In line 104, authors should briefly explain why they considered only preceding two weeks of depression and GAD as primary outcomes.

In line 106, authors should introduce a comma (,) after reference [15]

RESULTS

In line 127, authors should replace the word "participants" with the word "subjects"

Authors should convert some of their tabular data into graphical data. Readers would better appreciate graphical data than tabular data.

In line 156, authors should replace the word "particular" with the word "participants"

DISCUSSION

In line 164, authors should introduce a comma (,) after the word "face"

In line 167, should write the full meaning of GAD, then put the abbreviation in parenthesis before using it in subsequent sentences in lines 183 and 215. They should do same with the abbreviation "LMICS" as used in lines 181 and 234.

In line 170, authors should clearly state which participants they are referring to. I believe they are referring to recently diagnosed COVID-19 participants. This should be clearly stated.

In the sentence that begins with "The prevalence of depression..." in line 172, the results of both studies in the sentence are not comparable since both studies had different sample size. Authors should either remove this comparison or rephrase their sentence.

In the sentence that begins with "In the previous study..." in line 175, which previous study are authors referring to?

In line 189, authors should replace the preposition "by" with "of".

In line 193, authors should provide references after the phrase "psychological distress during the COVID-19 pandemic

In line 196, which counterparts are authors referring to? This should be well-defined.

In the sentence that begins with "Our study, is to our knowledge..." in lines 200 and 201, authors should rephrase the sentence.

In line 209, authors should introduce a hyphen (-) between "well" and "established". Thus, it should be written as "well-established"

Authors should provide references at the end of the sentence in line 210

LIMITATION

In line 223, the phrase "...and whether the association changed as time..." is confusing. Authors should rephrase it

In line 225, authors should provide references to the wider literature that corresponds to their finding. This should be followed by a comma (,)

CONCLUSION

In line 233, authors should replace the semi-colon (;) with a full stop (.). The word "therefore" that follows it should start with block letter "T".

REFERENCES

Authors should make sure that the format of their reference list aligns with the journal's requirement. They should also ensure that all in-text references are in the reference list and vice-versa.

6. PLOS authors have the option to publish the peer review history of their article (what does this mean?). If published, this will include your full peer review and any attached files.

Reviewer #1: No

Reviewer #2: **Yes: **Dr. George J. Dugbartey

---

## [Author Response · Author response to Decision Letter 0]

10 Apr 2023

Makerere University 

School of Public Health 

05.04.2023

The Academic Editor

PLOS ONE

We are grateful to the editor for giving us the opportunity to revise our manuscript titled " Association between recent COVID-19 diagnosis, depression and anxiety symptoms among slum residents in Kampala, Uganda. We also appreciate the peer reviewers' efforts for their insights which have significantly improved the quality of our manuscript. 

We have addressed the reviewers' comments to the best of our ability, hence would like to submit the revised manuscript.

Reviewer #1: Review comments

Title: Association between recent COVID-19 diagnosis on depression and anxiety symptoms among slum residents in Kampala, Uganda

Abstract

The background should mention briefly the rationale for choosing slum settlements.

Response: We have briefly explained that slums are associated with poor living conditions which are also associated with an elevated risk of mental health. We highlight that research on mental health among slum dwellers is limited but expected to be higher and with COVID-19, there was a need to understand this association further among slum dwellers. 

The tools that were used in data collection especially for anxiety and depressive symptoms should be mentioned.

Response: Thank you. We have highlighted the tools (PH-9 and GAD-7) used for data collection. We have written these in full the first time they appear in the manuscript. 

It will be important to indicate the incidence of covid-19 diagnosis within that past 30 days in the results

Response: Thank you. We have now reported on the prevalence of positive COVID-19 diagnosis instead of incidence as it was not possible to measure incidence with our study design. 

Introduction

Authors should provide references for sentence in lines 63 and 64

Response: Thank you for the suggestion. We have now provided these references. 

Generally, the introduction section is well written but authors should revise sections about low literature on COVID and slums since s cursory search online provides some evidence to guide the authors. Authors should focus building a case for their study based on what has been reported in relation to COVID and not so much on SARS etc.

Response: We have strengthened the introduction section by providing evidence in other countries in references 11 to 16. 

Methods

Location of the study

Some detailed description of the study area should be provided. What is the population, facilities, healthcare access, COVID related infection rates etc should be provided since it’s one of Kampala's most densely populated slums and such information should be readily available.

Response: Thank you for the comment. We have now updated this section with the information as suggested under the section on study design and setting. 

Sampling

Line 87 - Sample size was calculated for the main study. It’s not clear what authors mean though I guess it’s been described earlier?

Response: We have described that this is a secondary analysis. The main study has now been briefly described and a reference for more details has also been provided. 

Procedure

How was COVID diagnosis confirmed?

Response: We have now clarified this. COVID-19 diagnosis was self-reported by participants based on whether they had received a positive test in the previous 30 days or not from a hospital or laboratory. It was not confirmed as it was only self-reported.

How was the covid situation at the time of data collection and what considerations were made?

Response: A COVID-19 risk management protocol which was approved by the ethics committee was followed by the research team to minimize infection during data collection. This has now been included under the ethics section. At the time of data collection, the risk of COVID-19 in the country had decreased with low incidence. Nevertheless, the study team followed the appropriate guidelines during the process.

Ethics?

Information on covid protocol missing.

Response: We have included this under ethics. We have also provided an approved COVID-19 risk mitigation protocol used in our study in section on Ethical consideration … (on page 18).

Data collection and measures

The PHQ and GAD have to be described well. How were the responses scored and categorized based on the scores for the items in the questionnaire?

How did the authors ensure the validity/reliability of the questionnaire?

Was the study pilot tested? What was the outcome of the pilot and what modifications were made afterwards?

Response: We have briefly described the PHQ-9 and GAD-7 tools in section “Study procedures and measures”. on page …7. These are all validated tools and their reliability (Cronbach’s alpha) has now been provided which was high. Pretesting was done in another slum setting and a few modifications to the tools were made to improve clarity and understanding. This has been highlighted. 

Results

The results are well presented. A few comments though:

In the tables, there was information on substance use, income, covid experiences etc.

1. Each study variable has to be clearly indicated in the methods and how they were assessed.

Response: Thank you. We have now provided more information on these covariates including how they were measured and the rationale for categorizing some variables has been given under the methods sections of study procedures and data management.

2. What was the rationale for choosing the income categories?

Response: Thank you for the comment. These were based on median monthly incomes from the national report. A more detailed explanation for the choice of these categories is in the section on data management and analysis on page 8. 

Discussion.

I would rather authors stick to depressive and anxiety symptoms since the study tools and approach does not qualify for a diagnostic criterion for depression and anxiety to be used in the study.

Response: Thank you for your guidance. We have adjusted the wording as suggested. 

The strengths of this study did not come out well.

Response: Thanks for pointing this out. We have now stated key study strengths before the limitations. 

Conclusion

Present a brief summary of major findings and include the implications of such findings to practice, policy, clinical care and the current COVID-19 pandemic.

Response: Thank you for your guidance. We have improved the conclusion to reflect these suggestions. 

Reviewer #2: In this manuscript, Wafula et al aimed to establish a link between COVID-19 and depression and generalized anxiety disorder among poor people in inner-city slums in Uganda. While their finding is of interest, especially around this time when COVID-19 still prevails, the manuscript cannot be accepted for publication in its current form due to several issues including missing punctuation marks and use of wrong punctuation marks, which require a major revision. Below are my comments which authors should address to help improve the quality of their manuscript:

Response: Thank you for the commendation and the opportunity to revise this manuscript. We have endeavoured to address these concerns in the best way. 

TITLE

Authors should rephrase their title. They could either say "Association between recent COVID-19 diagnosis, depression and anxiety symptoms..." or "Effect of COVID-19 on depression and anxiety symptoms..."

Response: We have rephrased the title as “Association between recent COVID-19 diagnosis, depression and anxiety symptoms” as suggested. Thank you. 

ABSTRACT

Authors should write the full meaning of COVID-19, then put the abbreviation in parenthesis before using it in subsequent sentences. They should also write the full meaning of GAD. I believe it is generalized anxiety disorder. 

Response: Thank you for the suggestion which we have addressed.

They should specify whether the 284 individuals recruited in the study were children, adults or both. Otherwise, they should state the minimum age of the study participants.

Response: This has been clarified. These were adults (18 years and above).

 In addition, authors should also use p-values to state whether or not significant difference exists the study groups.

Response: We have now included the p-value as suggested. 

INTRODUCTION

In line 38, authors should write the full meaning of COVID-19, then put the abbreviation in parenthesis before using it in subsequent sentences.

Response: Thank you. We have done that as suggested.

Among the detrimental effects of COVID-19 in line 39, authors should include a statement on statistics of COVID-19-associated mortality.

Response: Thank you. We have provided the latest global statistics on morbidity and mortality. 

In line 46, authors should delete the preposition "of" that appeared after the word "comprise".

Response: Thank you. This has been deleted.

In line 61, authors should provide additional examples of the COVID-19-associated psychological distress beyond Burkina Faso, Ethiopia and Egypt.

Response: We have provided a few more evidence from different African countries. 

In line 71, authors should replace the word "elsewhere" with a word or a phrase that defines the opposite of slum dwellers or slum settlement. This should be followed by a cited reference from published literature.

Response: Thank you for the suggestion. We have made this change as suggested. 

In line 79, authors should replace semi-colon (;) with a comma (,)

Response: We have effected this change.

In line 80, authors should include a statement on why the study lasted from April to May, 2022

Response: Thank you but we didn’t think we needed to justify this because that is how long it took to complete our interviews/data collection phase. But for clarity, data collection started in April 2022 and ended in May 2022. 

Did authors consider any inclusion and exclusion criteria in their study design? If yes, what were these criteria? If no, why?

Response: We have now provided a brief eligibility criteria under subsection: Study design and setting.

In lines 86 and 87, authors should briefly state how they calculated sample size for the main study.

Response: We have provided additional details on the title of the study and the assumptions followed for the sample size calculation. 

In line 89, authors should briefly explain what they mean by "systematic sampling"

Response: We have provided the details regarding systematic sampling in the manuscript. 

What is the meaning of "LC" as used in line 91?

Response: We provided this in full at first use. LC stands for local council.

In line 92, authors should introduce a comma (,) after the word "direction"

Response: This change has been made. Thank you.

In line 104, authors should briefly explain why they considered only preceding two weeks of depression and GAD as primary outcomes.

Response: Thank you for the comment. This is the standard practice using the PHQ-9 and GAD-7 which all assess experiences over a 2-week period. 

In line 106, authors should introduce a comma (,) after reference [15]

Response: We have made this change. Thank you for the observation. 

RESULTS

In line 127, authors should replace the word "participants" with the word "subjects"

Response: Thank you for the suggestion. The change has been made. 

Authors should convert some of their tabular data into graphical data. Readers would better appreciate graphical data than tabular data.

Response: Thank you for the suggestion. However, we think that although graphs can improve comprehension, the available data does not bring the needed contrasts which would be great for the infographics. Due to this, we believe the table can still serve its purpose. 

In line 156, authors should replace the word "particular" with the word "participants"

Response: Thank you for your keen observation. We have changed as suggested. 

DISCUSSION

In line 164, authors should introduce a comma (,) after the word "face"

Response: Thank you. This change has been made. 

In line 167, should write the full meaning of GAD, then put the abbreviation in parenthesis before using it in subsequent sentences in lines 183 and 215. They should do same with the abbreviation "LMICS" as used in lines 181 and 234.

Response: Thank you for the observation. The suggested changes have been made.

In line 170, authors should clearly state which participants they are referring to. I believe they are referring to recently diagnosed COVID-19 participants. This should be clearly stated.

Response: Thank you. This is the general adult population regardless of whether they were recently diagnosed or not. It is important to note that only 11% of our participants had a recent COVID-19 diagnosis. Because of this, we decided to remain with the original word “participants”. 

In the sentence that begins with "The prevalence of depression..." in line 172, the results of both studies in the sentence are not comparable since both studies had different sample size. Authors should either remove this comparison or rephrase their sentence.

Response: Thank you for this statement. We have now rephrased the sentence to caution our readers of the fact that one is a larger national study in the general population and was not specifically focused on slum dwellers like our study was. However, we think the sample size difference should not discourage comparisons. 

In the sentence that begins with "In the previous study..." in line 175, which previous study are authors referring to?

Response: Thank you for pointing this out. We have clarified on the study we were referring to. 

In line 189, authors should replace the preposition "by" with "of".

Response: Thank you. We have changed this and many similar prepositions. 

In line 193, authors should provide references after the phrase “psychological distress during the COVID-19 pandemic

Response: Thank you for pointing this out. We have now inserted a number of references to support this statement.

In line 196, which counterparts are authors referring to? This should be well-defined.

Response: We meant “those who did not have a COVID-19 diagnosis within the previous 30 days” but we have now written it in full as recommended. 

In the sentence that begins with "Our study, is to our knowledge..." in lines 200 and 201, authors should rephrase the sentence.

Response: We have tweaked the sentence a bit to improve readability. Thank you for the observation. 

In line 209, authors should introduce a hyphen (-) between “well” and “established”. Thus, it should be written as “well-established”

Response: The hyphen has been introduced as suggested. 

Authors should provide references at the end of the sentence in line 210

Response: Thank you for the observation. We have now inserted the appropriate reference. 

LIMITATION

In line 223, the phrase "...and whether the association changed as time..." is confusing. Authors should rephrase it

Response: We have paraphrased this sentence as suggested and we believe it now reads better. 

In line 225, authors should provide references to the wider literature that corresponds to their finding. This should be followed by a comma (,).

Response: Thank you for the comment. The wider literature (references) is already compared and discussed earlier (as part of the discussion and introduction). We believe it is not necessary to cite this literature in the section on strengths and limitations. 

CONCLUSION

In line 233, authors should replace the semi-colon (;) with a full stop (.). The word "therefore" that follows it should start with block letter "T".

Response: Thank you. We have made changes to the overall conclusion and paid attention to the punctuation. 

REFERENCES

Authors should make sure that the format of their reference list aligns with the journal's requirement. They should also ensure that all in-text references are in the reference list and vice-versa.

Response: Thank you. We have checked and can confirm that indeed all intext citations are appearing in the list of references. 

Thank you, 

Sincerely, 

Solomon Wafula 

Corresponding Author

---

## [Decision Letter · Decision Letter 1]

20 Apr 2023

Association between recent COVID-19 diagnosis, depression and anxiety symptoms among slum residents in Kampala, Uganda

PONE-D-22-35388R1

Dear Dr. Wafula,

We’re pleased to inform you that your manuscript has been judged scientifically suitable for publication and will be formally accepted for publication once it meets all outstanding technical requirements.

Kind regards,

Seth Kwabena Amponsah, PhD

Academic Editor

PLOS ONE

Additional Editor Comments (optional):

Reviewers' comments:

Reviewer's Responses to Questions

**Comments to the Author**

1. If the authors have adequately addressed your comments raised in a previous round of review and you feel that this manuscript is now acceptable for publication, you may indicate that here to bypass the “Comments to the Author” section, enter your conflict of interest statement in the “Confidential to Editor” section, and submit your "Accept" recommendation.

Reviewer #1: All comments have been addressed

Reviewer #2: All comments have been addressed

2. Is the manuscript technically sound, and do the data support the conclusions?

Reviewer #1: Yes

Reviewer #2: Yes

3. Has the statistical analysis been performed appropriately and rigorously? 

Reviewer #1: Yes

Reviewer #2: Yes

4. Have the authors made all data underlying the findings in their manuscript fully available?

Reviewer #1: Yes

Reviewer #2: (No Response)

5. Is the manuscript presented in an intelligible fashion and written in standard English?

Reviewer #1: Yes

Reviewer #2: Yes

6. Review Comments to the Author

Reviewer #1: Authors have adequately addressed the comments raised.

They should consider revising the use of study subjects to study participants.

Thank you

Reviewer #2: The reviewer's comments were clearly answered, and appropriate changes were introduced in the revised manuscript.

7. PLOS authors have the option to publish the peer review history of their article (what does this mean?). If published, this will include your full peer review and any attached files.

Reviewer #1: No

Reviewer #2: **Yes: **George J. Dugbartey

---

## [Editor Report · Acceptance letter]

25 Apr 2023

PONE-D-22-35388R1 

Association between recent COVID-19 diagnosis, depression and anxiety symptoms among slum residents in Kampala, Uganda 

Dear Dr. Wafula:

I'm pleased to inform you that your manuscript has been deemed suitable for publication in PLOS ONE. Congratulations! Your manuscript is now with our production department. 

Kind regards, 

on behalf of

Dr. Seth Kwabena Amponsah 

Academic Editor

PLOS ONE